# Repetitive transcranial magnetic stimulation activates glial cells and inhibits neurogenesis after pneumococcal meningitis

Lukas Muri[1,2], Simone Oberhänsli[3], Michelle Buri[1,2], Ngoc Dung Le[1,2], Denis Grandgirard[1], Rémy Bruggmann[3], René M. Müri[4], Stephen L. Leib[1]*

1 Neuroinfection Laboratory, Institute for Infectious Diseases, University of Bern, Bern, Switzerland,
2 Graduate School for Cellular and Biomedical Sciences (GCB), University of Bern, Bern, Switzerland,
3 Interfaculty Bioinformatics Unit and SIB Swiss Institute of Bioinformatics, University of Bern, Bern, Switzerland, 4 Department of Neurology, University of Bern, Bern, Switzerland

* stephen.leib@ifik.unibe.ch

**Data Availability Statement:** Raw data for differential gene expression analysis are deposited at the European Nucleotide Archive (ENA,

## Abstract

Pneumococcal meningitis (PM) causes damage to the hippocampus, a brain structure critically involved in learning and memory. Hippocampal injury–which compromises neurofunctional outcome–occurs as apoptosis of progenitor cells and immature neurons of the hippocampal dentate granule cell layer thereby impairing the regenerative capacity of the hippocampal stem cell niche. Repetitive transcranial magnetic stimulation (rTMS) harbours the potential to modulate the proliferative activity of this neuronal stem cell niche. In this study, specific rTMS protocols–namely continuous and intermittent theta burst stimulation (cTBS and iTBS)–were applied on infant rats microbiologically cured from PM by five days of antibiotic treatment. Following two days of exposure to TBS, differential gene expression was analysed by whole transcriptome analysis using RNAseq. cTBS provoked a prominent effect in inducing differential gene expression in the cortex and the hippocampus, whereas iTBS only affect gene expression in the cortex. TBS induced polarisation of microglia and astrocytes towards an inflammatory phenotype, while reducing neurogenesis, neuroplasticity and regeneration. cTBS was further found to induce the release of pro-inflammatory cytokines *in vitro*. We conclude that cTBS intensified neuroinflammation after PM, which translated into increased release of pro-inflammatory mediators thereby inhibiting neuroregeneration.

## Introduction

During pneumococcal meningitis (PM), bacterial proliferation and autolysis in the cerebrospinal fluid (CSF) causes an excessive inflammatory reaction, which is associated with blood brain-barrier (BBB) breakdown, increased intracranial pressure, hydrocephalus and cerebral ischemia [1]. In patients with PM, cerebrovascular complication are frequently observed [2], with vasculitis and vasospasms being the cause for cerebral infarction and subsequent cortical damage during meningitis [1–3]. The occurrence of hippocampal apoptosis in the subgranular

accession number PRJEB37769). Additional relevant data are within the manuscript and R scripts used to assess differential gene expression are deposited as supporting information (S1 Supporting Information).

**Funding:** This work was supported by a grant from the Swiss National Science Foundation (Grant 189136 to SLL). The funder had no role in study design, data collection and analysis, decision to publish, or preparation of the manuscript.

**Competing interests:** The authors have declared that no competing interests exist.

zone of the dentate gyrus is a further hallmark of PM-induced neuronal damage [4–7]. Hippocampal apoptosis during PM occurs in recently postmitotic immature neurons in the dentate gyrus [7], thereby directly affecting the hippocampal stem cell niche and decreasing its capacity to regenerate [8]. Consequently, human observational studies repeatedly found long-term neurological sequelae after PM including sensorineural hearing loss, sensorimotor deficits, cognitive impairments and behavioural problems in up to 50% of survivors [2, 9, 10].

PM was shown to reduce hippocampal volume and decrease the amount of dentate granule cells [11]. Increasing damage of the hippocampus and its stem cell niche during PM has been experimentally shown to be associated with learning and memory deficits [12–14]. A transient increase in neuroproliferation has been observed after PM, but did not result in a net increase in dentate granule neurons, indicating that endogenous neurogenesis cannot compensate for hippocampal damage occurring during acute infection [11]. As PM-induced hippocampal apoptosis occurs in a region capable of endogenous neural self-renewal and repair, therapeutic strategies to support endogenous neuroregeneration after PM-induced brain injury represent a possible approach to improve the outcome of PM. In contrast to the relatively limited therapeutic window for effective prevention of brain injury during excessive PM-induced neuroinflammation, the potential to induce neuroregeneration by chronically applied substances might allow more time for therapeutic interventions [15]. Notably, dexamethasone–the only clinically recommended adjunctive therapy for PM–was repeatedly shown in experimental models to not only aggravate hippocampal apoptosis during acute infection [12, 16], but also decrease the neuroregenerative capacity of hippocampal stem cells *in vitro* and *in vivo* [17], thereby eventually causing a worse outcome in neurofunctional tests [12].

Repetitive transcranial magnetic stimulation (rTMS) is a non-invasive technique to induce local electric currents in the brain using the principle of electromagnetic induction [18]. Currently, rTMS is applied to treat many different neurological disease including addiction, stroke and depression [19–21], but its molecular effects remain elusive [21]. Recent experimental studies focusing on the molecular and cellular effects of rTMS found evidence for increased hippocampal neurogenesis and neuroplasticity upon stimulation [22–27]. This was associated with increased levels of brain-derived neurotrophic factor (BDNF) and increased activation of its receptor TrkB [22, 28], which may account for its beneficial effects as an anti-depressive therapy [26, 28]. Intermittent or continuous theta burst stimulation (iTBS or cTBS) represent specific and very potent rTMS protocols during which stimulations are applied as bursts of 3–5 pulses at 30–100 Hz repeated at 5 Hz, with iTBS lowering cortical excitability and cTBS enhancing it [29]. In experimental stroke, iTBS and cTBS were shown to upregulate the expression of genes involved in neuroplasticity, neuroprotection and cellular repair, eventually improving functional outcome [23]. As PM is associated with a decrease in hippocampal volume, a loss of dentate granule cells [11] and reduced neuroregenerative capacity [8], application of TBS may be used to induce neuroregeneration and to compensate for neural loss during PM, eventually improving the outcome after PM.

## Methods

### Infecting organism

A clinical isolate of *Streptococcus pneumoniae* (serotype 3) from a patient with bacterial meningitis was cultured overnight in brain heart infusion (BHI) medium, diluted 10-fold in fresh, pre-warmed BHI medium and grown for 5 h to reach the logarithmic phase. The bacteria were centrifuged for 10 min at 3100 x g at 4˚C, washed twice, resuspended in saline (NaCl 0.85%) and further diluted in saline to the desired optical density ($OD_{570nm}$). The inoculum concentration was determined by serial dilution and culturing on Colombia sheep blood agar (CSBA) plates.

## Infant rat model of pneumococcal meningitis and TBS protocols

All animal studies were approved by the Animal Care and Experimentation Committee of the Canton of Bern, Switzerland (license no. BE 129/14). A well-established infant rat model of pneumococcal meningitis was used for this study [30, 31]. Eleven-day old male Wistar rat pups and their dam were purchased from Charles Rivers (Sulzfeld, Germany). The dam was provided with tap water and pellet diet at libitum. Litter was kept in a room at a controlled temperature of $22 \pm 2°C$ and with natural light.

Infant rats were infected by intracisternal injection of 10 μl bacterial inoculum containing $4.2 \times 10^5$ CFU/ml of living *S. pneumoniae*. Meningitis was confirmed by quantitative analysis of bacterial titres in the cerebrospinal fluid (CSF) at 18 h post infection (hpi), where 5 μl of CSF were collected by puncture of the cisterna magna, followed by serial dilution and cultivation on CSBA plates. A total of 12 infant rats were included in this study. All animals received antibiotic therapy consisting of ceftriaxone (100mg/kg, i.p. twice daily, Rocephin, Roche) started at 18 hpi and continued for 5 days. Animals were weighted and clinically scored according to the following scoring scheme (1 = coma, 2 = does not turn upright, 3 = turns upright in > 5 s, 4 = turns upright in < 5s, 5 = normal) at 0, 18, 24 and at 42 hpi. Five days after infection–when the acute phase of infection with associated neuroinflammation is overcome and animals start to recover from pneumococcal meningitis–animals were randomized to receive continuous theta burst stimulation (cTBS, n = 4; three 30 Hz pulses repeated at intervals of 100 ms for 200 times), intermittent theta burst stimulation (iTBS, n = 4, ten 50 Hz bursts with 3 pulses each repeated 20 times at 5 Hz intervals) or sham stimulation (sham, n = 4) on two consecutive days with four stimulations à 600 pulses per day (0min, 15min, 60min, 75min), resulting in a total of 4800 pulses. TBS was applied using a Cool-40 rat coil (MagVenture, Denmark) with 16% output intensity, representing 90% stimulation intensity of the previously assessed motor threshold of 18%. During stimulation protocols, animals were un-anesthetized but restrained in a commercially available conic plastic bag, with the nose outside the bag to allow breathing. To reduce stress related to restraining, animals were familiarized with the bag and the restraining for two days before stimulation. Twenty-four hours after the last stimulation, animals were sacrificed with an overdose of pentobarbital (150 mg/kg, i.p., Esconarkon, Streuli Pharma AG, Switzerland) and perfused with phosphate-buffered saline (PBS). Cortical and hippocampal tissue from the left hemisphere was harvested, frozen immediately on dry ice and kept at -80°C for RNA isolation. The right hemisphere was fixed in 4% PFA for 4 h and stored in PBS at 4°C for embedding in paraffin.

## RNA isolation

RNA isolation was performed as previously described [32, 33]. In brief, frozen tissue (hippocampus or cortex) from the left hemisphere were put into 1 ml QIAzol Lysis reagent (Qiagen, Hilden, Germany) and immediately homogenized by a rotor-stator homogenizer (TissueRuptor, Qiagen, Hilden Germany). RNA was isolated using the RNeasy Lipid Tissue Mini Kit (Qiagen), following the manufacturers protocol. To remove contaminating DNA, 20 μl isolated RNA was treated with DNase using the DNA-free Kit (Ambion, Carlsbad, CA, USA). RNA quality and quantity were determined on the Agilent 2100 Bioanalyzer platform (RNA 6000 Nano, Agilent Technologies, Waldbronn, Germany) and validated on the NanoDrop device (NanoDrop, Wilmington, USA).

## Whole transcriptome analysis and RNAseq

Sequencing data were generated by the NGS Platform of the University of Bern. Differential gene expression analysis was performed by the Interfaculty Bioinformatics Unit (IBU) of the

University of Bern. Samples were sequences with Illumina (TruSeq® Stranded mRNA Library Prep, single reads of 100bp length, Illumina®, San Diego, USA). Between 26 and 48 million reads were obtained per sample. The quality of the RNA-seq data was assessed using fastqc v. 0.11.5 [34]. Reads were mapped to the reference genome (Rnorvegicus.6.0) using HiSat2 v. 2.1.0 [35]. FeatureCounts v. 1.6.0 was used to count the number of reads overlapping with each gene as specified in the genome annotation (Rnor_6.0.92) [36]. The Bioconductor package DESeq2 v. 1.18.1 was used to test for differential gene expression between the experimental groups [37]. All analyses were run in R v. 3.4.4. Gene set enrichment analysis was performed with differentially expressed genes comparing TBS stimulated and sham-treated animals using Gene Ontology enRIchment anaLysis and visuaLizAtion tool (GOrilla). Genes with a Benjamini-Hochberg adjusted p-value < 0.1 (representing raw p-values < ~0.005) were included in the analysis for differentially regulated gene ontologies.

Raw data has been deposited at the European Nucleotide Archive ENA (accession number PRJEB37769).

## Immunofluorescence analysis quantifying CD68+ cells in the cortex and hippocampal dentate gyrus

Right brain hemispheres embedded in paraffin were cut to 10 μm sections using a microtome (Microm, Germany). Every 18th section was sampled on Superfrost Plus Menzel glass slides and air dried. Sections were deparaffinized and submitted to antigen retrieval by incubating the slides in sodium citrate (Merck KGaA) 10mM pH 6.0 for 1 h in a 95˚C water bath. Sections were permeabilized for 5 min with 0.1% Triton-X, followed by blocking with blocking solution (PBS with 2% BSA and 0.01% Triton-X) for 1 h at room temperature (RT). The primary antibodies against CD68 (1:500, mouse-anti-rat, Bio-Rad/Serotec, MCA341R) and Iba1 (1:500, rabbit polyclonal, FUJIFILM Wako Chemicals, WA3 019–19741) were diluted in blocking solution, added to the slides and incubated overnight at 4˚C. The slides were washed 3 x 5 min with PBS and the secondary antibodies–donkey-anti-mouse Cy3 (1:500, Jackson ImmunoResearch, 715-165-151) and goat-anti-rabbit Alexa Fluor 488 (1:500, Thermo Fisher Scientific, A11034)–were added for 2 h at RT. After washing the sections 3 x 5 min in PBS, they were mounted with Fluoroshield containing DAPI and kept at 4˚C in dark until imaging. CD68+ region of the cortex and the hippocampal dentate gyrus were quantified using a x200 magnification on fluorescent microscope. The first 3 sampled sections containing the dentate gyrus with the upper and lower blades connected–when cutting coronal sections from anterior to posterior–were quantified. Mosaic pictures of each dentate gyrus were created with Zeiss Axio-Vision software using individual pictures taken at x200 magnification. Cortical regions focusing on a 3x3 mosaic (x200 magnification) dorsal to the assessed dentate gyrus were systematically assigned for evaluation. Areas of cortical and hippocampal tissue was evaluated using ImageJ software. Number of CD68+ cells were calculated and evaluated as number of CD68+ cells per mm$^2$ cortex or dentate gyrus granule cell layer.

## *In vitro* astroglial cells stimulation and cytokine release

Astroglial cells were isolated from the brains of infant Wistar rats at postnatal day 3 (P3) received from the Central Animal Facility of the Department for BioMedical Research of the University of Bern–as previously described [38, 39]. The rats were sacrificed by decapitation and brains were isolated. The cortices were homogenized mechanically in PBS by pipetting up and down with a 5mL plastic pipette, centrifuged (500 x g, 7 min, 4˚C) and resuspended in DMEM (Sigma-Aldrich, Merck, Switzerland) containing 5% FCS (Biochrom, Germany), GlutaMAX™ (ThermoFisher, Switzerland) and antibiotic-antimycotic solution (ThermoFisher,

Switzerland). After resuspension, cells were plated in T75 flask (TPP®, Merck, Switzerland) previously coated with poly-L-ornithine (PLO, 0.01 mg/ml in PBS, Sigma-Aldrich, Merck, Switzerland) for 4 h. On day 11 post isolation, cells were seeded on PLO-coated 24 well plates, at a density of 200'000 cells/well. Astroglial cells were stimulated with 4x cTBS or sham-stimulation on two consecutive days, representing the same stimulation procedure as *in vivo*. Cytokine release was assessed using magnetic multiplex assay (Rat Magnetic Luminex® Assay, Rat Premixed Multi-Analyte Kit, R&D Systems, Bio-Techne, R&D Systems Inc., USA) on a Bio-Plex 200 station (Bio-Rad Laboratories, Germany) as described previously [39–41], where 50 μl of undiluted cell culture medium was used. For each sample, a minimum of 50 beads was measured.

## Statistical analysis

Statistics used to assess differential gene expression are mentioned above. R scripts used to assess differential gene expression are deposited as (S1 File). Statistical analyses for assessing differences in bacterial titres, weight change during acute PM, CD68+ cell quantification and *in vitro* cytokine analysis were performed with GraphPad Prism (Prism 7; GraphPad Software Inc., San Diego, USA). Results are presented as mean values ± standard deviation (SD) if not stated otherwise. To compare differences between means of two normally distributed groups, an unpaired Student *t* test was used. A two-way ANOVA was performed to analyse differences of weight change over time. For CD68+ cell quantification and bacterial titre evaluation comparing multiple different groups, a Tukey's multiple comparison test was applied to adjust for multiple testing. A *p* value of $< 0.05$ was considered statistically significant with with $p<0.05$ (*) and $p<0.01$ (**).

## Results

All 12 infant rats enrolled in the *in vivo* study developed pneumococcal meningitis after intracisternal injection of *S. pneumoniae* proven by bacterial growth in CSF samples obtained at 18 hpi ($>10^7$ CFU/ml), reduced clinical scores and weight loss. Bacterial CSF titres were comparable between the three different groups (S1 Fig panel A). After infection, no statistically significant difference between sham and iTBS (2-way ANOVA F = 0.2059 and p = 0.6524 with degree of freedom (df) = 1; interaction of main factors p = 0.9866) or sham and cTBS (2-way ANOVA F = 0.0004 and p = 0.9847 with df = 1; interaction of main factors p = 0.9999) in terms of weight development was detectable (S1 Fig panel B). Together, these data indicate that animals at time of exposure to TBS had experienced PM with a comparable degree of severity.

### Changes in gene expression upon TBS

Cortical and hippocampal whole transcriptome analysis after recovery from pneumococcal meningitis followed by two consecutive days of TBS revealed significant differences between cortical and hippocampal tissues with comparably less distinction between the different TBS protocols as revealed by principle component analysis (PCA, Fig 1).

Nevertheless, cTBS induced differential gene expression in 1308 genes in cortical tissue and 495 genes in hippocampal tissue compared to sham stimulation while iTBS only affected the expression of 69 genes in the cortex and did not induce any significant gene expression changes in the hippocampus. PCA of the 500 most variable genes, however, revealed no clear clustering of cTBS treated infant rats for cortical tissue. In hippocampal samples, PCA showed a clustering of cTBS treated animals distinctive from iTBS and sham-treated animals (Fig 2).

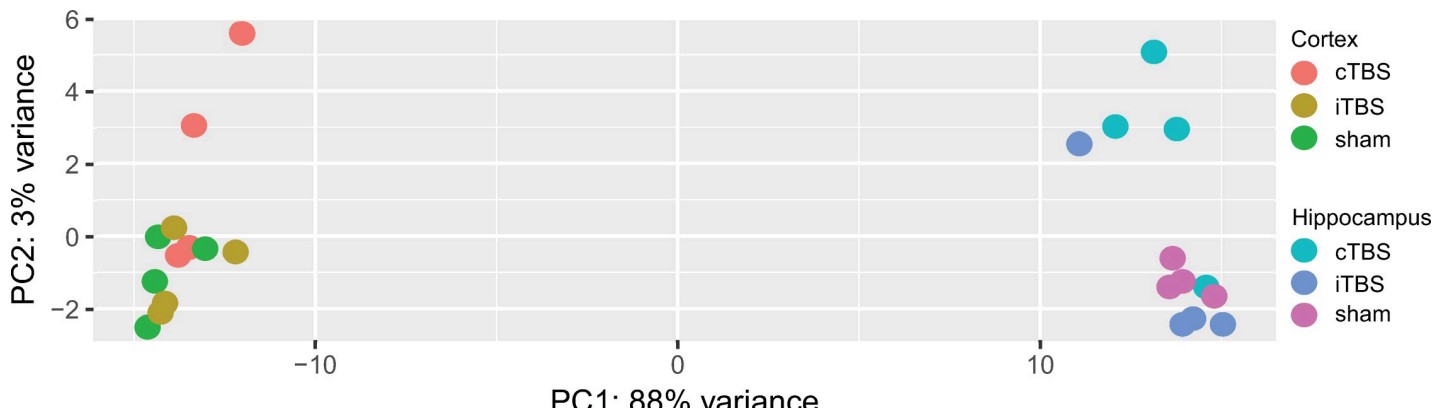

**Fig 1. Principal component analysis (PCA) including the 500 most differentially expressed genes comparing cortical and hippocampal tissue and different TBS protocols.** Two dimensional PCA plot revealed that the two different tissues (hippocampus in purple-blues and cortex in green-orange-red) account for the greatest variance within the samples, whereas the within-tissue difference by TBS protocols was comparably low.

Gene ontology (GO) analysis shed insight into differentially expressed biological processes upon TBS. In the cortex and compared to sham-stimulated animals, cTBS caused an upregulation of genes involved in 1.) metabolic processes with upregulated translation, amide biosynthesis and RNA processing; and 2.) response to stimuli with upregulated immune system processes, defence response and positive regulation of TNF-α response (S2 Fig). On the other side, cTBS downregulated gene expression related to 1.) regulation of central nervous system development with downregulated neurogenesis, axonogenesis and neuron projection

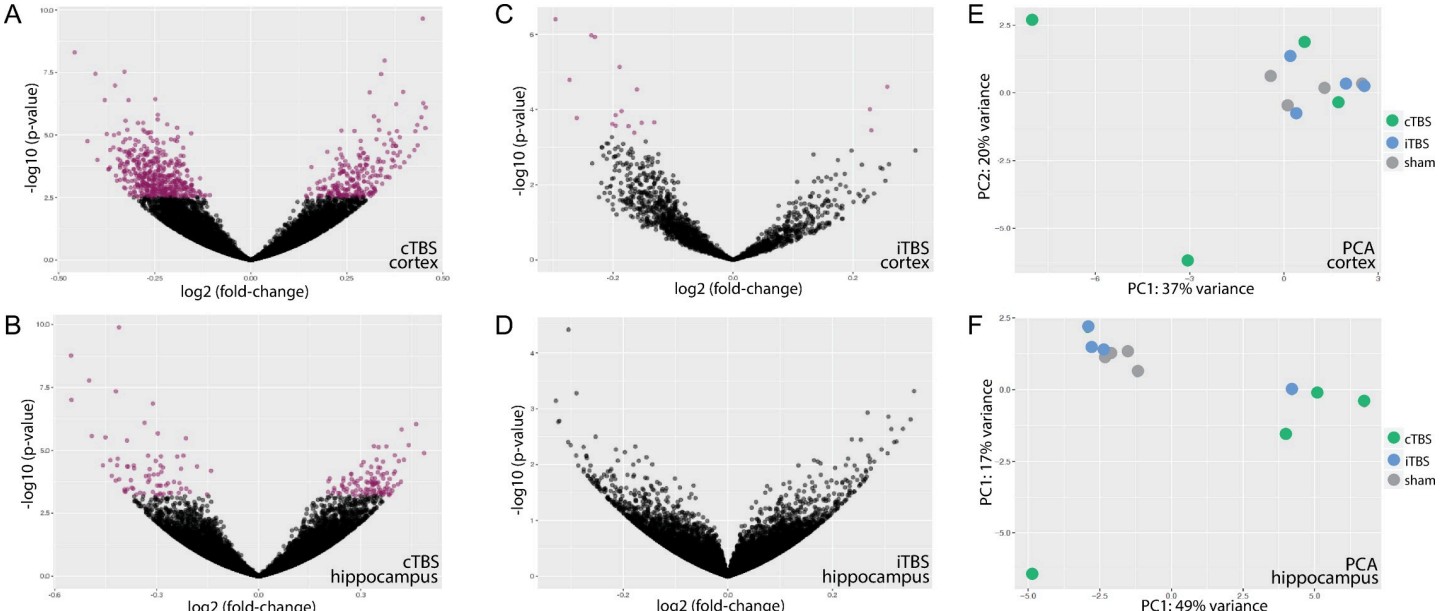

**Fig 2. Differential gene expression in TBS-treated animals compared to sham-treated animals in cortical and hippocampal tissue depicted in volcano plots and summarised in two dimensional PCA plots.** cTBS clearly induced differential up- and downregulation of cortical genes (**A**), an effect less pronounced in the hippocampus (**B**). iTBS only marginally affected cortical gene expression (**C**) and did not induce any differential expression of hippocampal genes (**D**). Two dimensional PCA plots according to the 500 most differential expressed gene did not reveal a clear clustering in cortical samples (**E**) but shows a cluster of cTBS-treated animals different from sham- or iTBS-treated animals in hippocampal samples (**F**). Purple dots in volcano plots represent differentially expressed genes with p-values <0.5 after Bejamini-Hochberg adjustment for multiple testing.

morphogenesis; 2.) regulation of signalling with downregulated modulation of chemical synaptic transmission and downregulated regulation of excitatory postsynaptic potentials; 3.) regulation of transport with downregulated ion transmembrane transport; and 4.) nervous system processes with downregulated learning, memory and cognition (S3 Fig).

Similarly, in the hippocampus and compared to mock-stimulated animals, cTBS induced an upregulation of 1.) metabolic processes with increased translation and ribosomal subunit assembly; and 2.) response to stimuli with increased cellular response to hydrogen peroxide (S4 Fig). Negatively regulated biological processes in the hippocampus after cTBS involed 1.) negative regulators of transcription and translation; 2.) cellular response to brain-derived neurotrophic factor (BDNF); 3.) regulation of transforming growth factor beta receptor signalling pathway; and 4.) cellular development processes with decreased neuron differentiation (S5 Fig).

Notably, as less genes were differentially regulated in the hippocampus compared to the cortex, the p-value for differentially regulated GO after adjustment for multiple testing by the Benjamini and Hochberg method, were higher with less GO reaching statistical significance.

The 60 downregulated cortical genes after iTBS compared to mock-stimulated animals were involved in 1.) regulation of trans-synaptic signalling with downregulated synaptic plasticity; 2.) nervous system processes with downregulated cognition, learning and memory; 3.) cell communication with downregulated chemical synaptic transmission and downmodulated neurotransmitter transport; and 4.) regulation of developmental processes with downregulated nervous system development, synapse maturation, neuron projection development and dendritic spine development.

Of note, as only 60 genes were differentially downmodulated by iTBS, GOs representing regulation of developmental processes were not statistically significant after adjustment for multiple testing. As 3 of the 9 upregulated genes in the cortex upon iTBS are involved in translation, GOs representing peptide biosynthetic processes and translation were upregulated but without statistical significance after adjustment for multiple testing. Since iTBS did not induce any differential gene expression in the hippocampus, no GOs could be analysed. Tables including a complete list of down- and upregulated GO, are found in the appendix (S1–S6 Tables).

## Upregulation of glia markers

Focussed analysis on differentially expressed glial marker revealed a significant upregulation of typical microglial activation markers upon cTBS compared to sham-stimulation (Table 1). Considering raw p-values of differentially expressed gene, the classical microglia activation marker *Iba1*, *Cd14*, *Cd45*, *Cd68*, *Cd86* and *F4/80* were found to be upregulated. In addition, markers for immune response such as toll-like receptors (*Tlr2*, *-3*, *-4* and *-7*), complement components (*C1qa*) and cytokine receptors (*Il-1r1*) were upregulated. Markers of astrocyte activation, namely *Gfap*, *S100b* and *S100a4* were significantly upregulated or showed trends towards increased expression. After adjusting for multiple testing, *Iba1*, *Tlr7*, *C1qa*, *Cd84* and *S100a4* kept statistical significance for increased expression after cTBS. On the other hand, a marker associated with alternative (M2) microglial activation (*CD163*) showed a trend for decreased expression, and other specific M2 markers (*Arg1*, *Cd206*, *Ym1*, *Fizz1*) were not differentially expressed. Suppressor of cytokine signalling 7 (*Socs7*) was downregulated, but without statistical significance after adjustment for multiple testing.

In general, the changes in expression levels remained relatively small comparing the different stimulation protocols (max. log2 fold change range from -0.6 to +0.5). However, the whole cortical or hippocampal tissue was analysed here. The signal due to genes specifically expressed by a given cell population (i.e. microglia) may have therefore been decisively diluted.

**Table 1. Differential regulation of cortical genes involved in microglia and astrocyte activation upon cTBS compared to sham-stimulation.**

| Gene | Log2 fold change | Raw p-value | Adjusted p-value |
|---|---|---|---|
| Iba1 | 0.346 | 0.00012 | 0.00853 |
| Cd68 | 0.358 | 0.00008 | n/a |
| Cd14 | 0.252 | 0.00565 | n/a |
| Ptprc (CD45) | 0,272 | 0.00863 | 0.08576 |
| Adgre1 (F4/80) | 0,295 | 0.00469 | 0.06291 |
| Tlr2 | 0.276 | 0.00302 | 0.05001 |
| Tlr3 | 0,299 | 0.00405 | 0.05962 |
| Tlr4 | 0.220 | 0.03398 | n/a |
| Tlr7 | 0,317 | 0.00121 | 0.03053 |
| C1qa | 0,320 | 0.00199 | 0.04056 |
| Il1r1 | 0.189 | 0.03875 | 0.19240 |
| Cd84 | 0.361 | 0.00050 | 0.01810 |
| Cd86 | 0.202 | 0.03679 | n/a |
| Cd163 | -0.179 | 0.05255 | n/a |
| Socs7 | -0.246 | 0.00737 | 0.07917 |
| Gfap | 0.178 | 0.06044 | 0.24493 |
| S100b | 0.206 | 0.00849 | 0.08497 |
| S100a4 | 0.434 | 0.00003 | 0.00445 |

Many genes generally associated with microglial activation (*Iba1*, *Cd68*, *Cd14*, *CD45*, *F4/80*) were upregulated after cTBS with low p-values ($p<0.01$). Genes associated with anti-inflammatory effects of microglia (*Cd163*, *Socs7*) showed trends for negative regulation. Genes classically linked to astrocyte activation (*Gfap*, *S100b*, *S100a4*) were either upregulated or showed trends for increased expression. Adjusted p-values with the value "n/a" may arise after automatic independent filtering for low mean normalised counts during differential gene expression analysis with DESeq2.

## cTBS increases abundance of CD68+ cells in cerebral cortex and hippocampal dentate gyrus

Immunofluorescence analysis of CD68[+] cells in the cortex and hippocampal dentate gyrus revealed that cTBS–but not iTBS–increased the abundance of CD68[+] cells in both assessed tissues (Fig 3). After adjustment for multiple testing, exposure to cTBS on two consecutive days resulted in a significant higher number of CD68[+] cells in the cortex compared to iTBS (76.61 vs. 13.79 cells/mm$^2$ tissue, $p = 0.0299$) and compared to sham stimulation (76.61 vs. 13.68 cells/mm$^2$ tissue, $p = 0.0205$, Fig 3D). In the granular cell layer of the hippocampal dentate gyrus, cTBS increased the abundance of CD68[+] cells compared to iTBS (134.6 vs. 63.02 cells/mm$^2$ tissue, $p = 0.0165$) and compared to sham stimulation (134.6 vs. 33.39 cells/mm$^2$ tissue, $p = 0.0014$, Fig 3C).

## Astroglial cell activation *in vitro*

Neonatal rat astroglial cells–containing astrocytes, microglia and oligodendrocytes–kept in culture for two weeks after isolation were exposed to 4 trains of cTBS or sham-stimulation on two consecutive days. Stimulation with cTBS only marginally affected the release of inflammatory cytokines in the cell culture supernatant (S6 Fig). Nevertheless, cTBS significantly increased the release of IL-1β ($p = 0.0163$), IL-10 ($p = 0.0327$) and TNF-α ($p = 0.0119$) compared to sham-stimulation. cTBS also increased levels of IL-6, but only with a statistical trend ($p = 0.1036$).

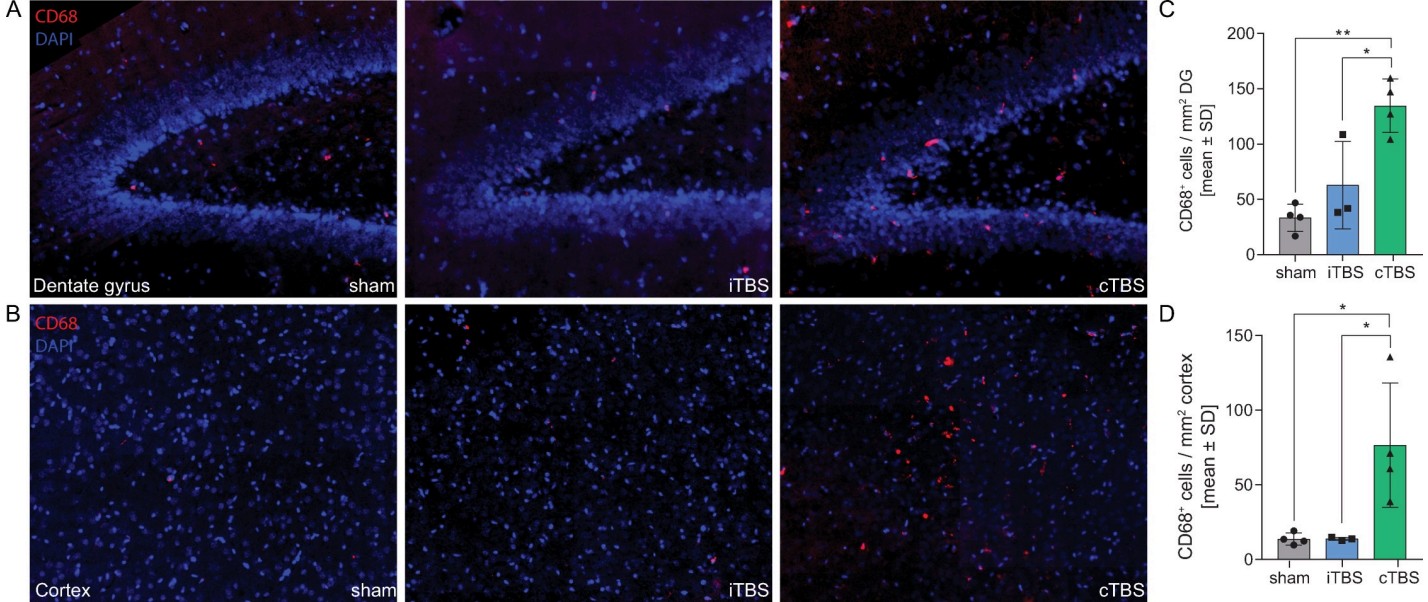

**Fig 3. cTBS increases abundance of CD68+ cells in cerebral cortex and hippocampal dentate gyrus.** Exposure to cTBS–but not to iTBS–increased the number of CD68+ cells in the granular cell layer of the hippocampal dentate gyrus (DG) compared to sham-stimulated and iTBS-stimulated animals (**A,C**). Similarly, cTBS increased the abundance of CD68+ cells in the cerebral cortex (**B,D**). Statistical differences were assessed using one-way ANOVA with Tukey's multiple comparison test to adjust for multiple testing.

## Discussion

In the present study, we studied the impact of TBS in infant rats microbiologically cured from PM after five days of antibiotic therapy. To ensure that magnetic stimulation does not interfere with neuroinflammatory processes during acute infection, stimulation was initiated at day five after infection. Previous studies revealed that at this time animals successfully survived PM and start to recover with neuroinflammatory markers and clinical scores being back at baseline and animals starting to gain weight again [38, 39, 41–44]. Our data suggest that, at the level of gene expression, TBS inhibited processes related to cortical and hippocampal neurogenesis, neurodifferentiation and neuroplasticity assessed 24 hours after the last stimulation trains. This effect was stronger after the cTBS protocol and more prominently affected gene expression in the cortex than in the hippocampus, where cTBS induced a downregulation of neurogenesis processes, synaptic transmission and plasticity as well as processes involved in cognition, learning and memory. Stimulation with the iTBS induced a less pronounced effect which was only detectable in the cortex. iTBS repressed gene expression pathways related to neuroplasticity and cognition, learning and memory. Both stimulation protocols induced an upregulation of translational processes, while cTBS specifically upregulated processes participating in immune system regulation and defence response. As both TBS protocols showed higher numbers of differentially expressed genes in the cortex compared to the hippocampus, we propose that the strength of magnetic fields might be attenuated in deeper structures, thus inducing a less pronounced effect in the hippocampus.

The upregulation of inflammatory gene ontologies suggested an involvement of microglia and astrocyte activation. Focussed analysis revealed that markers of pro-inflammatory microglia (M1 phenotype) [45–47] (*Iba1*, *Cd14*, *Cd45*, *Cd68*, *Cd84*, *F4/80*, *Tlr2, -4, -7, Il1r1* and *Cd86*) were upregulated, whereas markers for the microglial M2 phenotype (*Cd163*, *Arg1*, *Cd206*, *Ym1 and Fizz1)* showed a trend for downregulation or were not affected in cTBS

treated cortices compared to mock-stimulation. Gene expression in the hippocampus was less prominently altered. Nevertheless, *Iba1*, *Tlr2* and *Cd68* were also significantly upregulated in the hippocampus after cTBS, pointing towards a classical M1 polarisation of microglia. On the other hand, M2 markers in the hippocampus were unaffected. Immunofluorescence analysis validated findings from gene expression data. Quantification of CD68$^+$ signals–a marker for microglial activation with phagocytic activity [48, 49]–revealed an increased abundance of activated microglia in the cerebral cortex and the subgranular cell layer in the hippocampal dentate gyrus (Fig 3A–3C). The universal astrocyte marker *Gfap* and *S100b* showed a trend to be upregulated in cortices of cTBS stimulated rats, without reaching statistical significance. *S100a4*, which is associated with astrocyte migration and astrocyte activation upon neuronal damage [50, 51], was significantly upregulated.

Together these data indicate a M1 polarisation of microglia and an activation of astrocytes upon TBS after PM. This is in line with previous studies, where rTMS was shown to induce glia cell activation (reviewed by Cullen and Young, 2016 [52]). Magnetic stimulation of cultured astrocytes was shown to transiently increase protein levels GFAP [53]. High-frequency rTMS dramatically increased *Gfap* mRNA levels in the hippocampus and cortex in healthy mice [54]. In a rat model of demyelinating spinal cord injury, rTMS increased *Gfap* expression with increasing magnetic stimulation frequency [55], and increased astrocyte migration to lesion area [55, 56]. Furthermore, rTMS is reported to induce a upregulation of GFAP and IBA1 immunoreactivity in a gerbil model of ischemia, when magnetic stimulation was initiated immediately after ischemic injury [57]. This model of experimental ischemia in gerbils with early TMS treatment [57] resembles our pneumococcal meningitis model characterized by an intense neuroinflammation upon infection and focal ischemia. We found increased *Iba1* and *Gfap* (without reaching statistical significance after adjustment for multiple testing) expression levels pointing toward increased glia activation after exposure to cTBS. Based on whole tissue transcriptomic analysis, our data cannot discriminate between activated microglia and recruited macrophages/monocytes. However, as cTBS treatment was started five days after infection–when peripheral inflammatory cells should have been cleared from the CNS and inflammatory mediators are reported to be reduced to baseline [58]–we suggest that increased immune response markers and M1 polarisation markers derive from activated microglia and not from recruited macrophages. Generally, the effect of rTMS on astrocytes and microglia is reported to be context-dependent and therefore relies on the cellular environment [52]. This may explain our *in vitro* data showing rather small effects of rTMS on non-activated, healthy astroglial cell cultures, as reported for healthy animals [59]. Yet another explanation for increased neuroinflammation *in vivo* might arise from rTMS-induced changes in BBB permeability. High-intensity magnetic stimulation harbours the potential to increase BBB permeability most likely via increased release of glutamate [60]. Increased brain endothelial permeability to serum proteins was found to be associated with neuroinflammatory markers such as astrocyte transformation showing increased GFAP levels [61–64]. Changes in BBB permeability upon TBS and direct activation of glial cells might both contribute to the observed neuroinflammatory reaction upon magnetic stimulation. More experiments are needed to confirm these hypotheses.

Our observed data from an experimental model of pneumococcal meningitis suggest that glial activation during neuroinflammation is further intensified by TBS protocols leading to increased expression of astrocyte and microglia M1 markers, thereby aggravating or prolonging neuroinflammation with a negative impact on neuroregenerative mechanisms. This is in contrast to recent studies, where rTMS was associated with increased levels of BDNF and activation of its receptor TrkB [22–28] and increased hippocampal neurogenesis and neuroplasticity, resulting in beneficial effects during anti-depressive therapy [26, 28]. In an experimental stroke

model, iTBS and cTBS were neuroprotective and increased neuroplasticity and cellular repair, eventually leading to improved functional outcome [23]. There are, however, crucial differences between our study setting and previously reported studies reporting increased neurogenesis. We applied a comparably short stimulation protocol with four stimulation trains per day on 2 consecutive days, which represents a setup that showed beneficial effect in treatment of human neglect patients after stroke [65–67]. This is in contrast to stimulation for up to 60 days in studies with reported positive effect on neurogenesis [23, 68, 69]. In addition, we isolated brain tissues 24 hours after the last stimulation train to assess short-term effects of TBS, compared to some studies with significant longer poststimulation assessment [56]. Timing and duration of the stimulation protocol is critical and may reverse an improved outcome with increased neurogenesis to reduced neuroregenerative capacity. Furthermonre, studies reporting increased neurogenesis with increased neural stem cell progenitor proliferation, were often performed in healthy rodent models in the absence of neuroinflammation [25, 26, 70]. Additionally, induced neuroregeneration upon rTMS therapy was mostly found in adult rodent models [23, 25, 70]. These results might therefore differ from our infant model, where the rate of endogenous neurogenesis is already considerably high [11]. In line with our findings, rTMS treatment did not reverse the suppressed proliferation and survival of newly generated hippocampal granule cells observed in rats submitted to chronic psychosocial stress induced by social defeat but even further suppressed the survival rate of hippocampal neural stem cells [71].

Activated microglia were repeatedly reported to inhibit neurogenesis. In experimental pneumococcal meningitis, M1-polarised microglia (expressing inducible nitric oxide synthase, iNOS) were prominently found in the neurogenic niche of the hippocampal [11]. Treatment with a specific iNOS inhibitor restored neurogenesis, demonstrating the link between activated microglia, secreted nitric oxide and reduced neurogenesis [11]. In experimental LPS-induced neuroinflammation in rats, a negative correlation between activated microglia and the number of surviving hippocampal progenitor cells was found, indicating that activated microglia strongly impair hippocampal neurogenesis [72, 73]. In this model, selective microglia inhibition by minocycline was shown to restore neurogenesis [72]. Inflammatory mediators like IL-1β, TNF-α, IL-6 and nitric oxide are reported to be responsible for the negative effects of activated microglia on newly generated neurons [72]. Based on the results generated within the present study, where we found increased microglia and astrocyte activation and reduced neurogenesis and neuroplasticity upon TBS therapy, we suggest that neuroinflammation after PM is further intensified by TBS (especially cTBS), leading to increased release of microglial inflammatory mediators with deleterious effects on newly proliferated neurons, eventually reducing neurogenesis and neuroregeneration.

Our study has some limitations, including small numbers of animals in the *in vivo* experiments and the lack of an uninfected control group receiving the same stimulation procedures. As we were primarily interested in investigating the effects of TBS after PM and did not intent to focus on differences in gene expression between infected and non-infected animals, this control group was omitted. The effect of TBS on differential gene expression in healthy animals would represent another story. Furthermore, it would have been interesting to also analyse the effect of long-term TBS exposure on neurogenesis and neuroinflammation. However, this would have been beyond the scope of this study focussed on differential gene expression induced by TBS after acute PM.

## Conclusion

In the recovery phase of PM (i.e. 5 days after infection), cTBS stimulation for two consecutive days induced microglial M1-polarisation and astrocyte activation in the cortex and

hippocampus of infant rats. TBS induced a downregulation of genes related to neurogenesis, neuroplasticity and nervous system processes associated with cognition, learning and memory, most likely as a consequence of accentuated neuroinflammation. We conclude that TBS–especially cTBS–after PM is detrimental for the disease outcome as it increases neuroinflammation and reduces neuroregeneration.

## Supporting information

**S1 Fig. Bacterial CSF titres and weight development during pneumococcal meningitis.** Bacterial CSF titers were comparable between the three different groups cTBS, iTBS and sham and indicated a comparable severity of infection (A). Development of relative weight after infection further proved comparability between different groups, as PM-induced weight loss and recovery was non-different within all analysed animals analysed by 2-way ANOVA (B). Statistical differences for bacterial titres were assessed using one-way ANOVA with Tukey's multiple comparison test to adjust for multiple testing.
(PDF)

**S2 Fig. Overrepresented gene ontologies in upregulated genes upon cTBS in the cortex.** Significantly overrepresented gene ontologies are highlighted in grey, with darker colour representing more significant p-values (see scale). In the cortex, cTBS induced a significant upregulation of processes involved 1.) translation, 2.) RNA processing, 3.) ribonucleoprotein complex assembly, 4.) response to stimuli, 5.) immune system regulation and 6.) regulation of TNF production.
(PDF)

**S3 Fig. Overrepresented gene ontologies in downregulated genes upon cTBS in the cortex.** Significantly overrepresented gene ontologies are highlighted in grey, with darker colour representing more significant p-values (see scale). In the cortex, cTBS induced a significant downregulation of processes involved 1.) regulation of transmembrane transport, 2.) regulation of developmental processes including neurogenesis and axonogenesis, 3.) regulation of synaptic plasticity and excitatory postsynaptic potentials, 4.) neuron projection morphogenesis, 5.) transmembrane ion transport and 6.) nervous system processes such as cognition, learning and memory.
(PDF)

**S4 Fig. Overrepresented gene ontologies in upregulated genes upon cTBS in the hippocampus.** Significantly overrepresented gene ontologies are highlighted in grey, with darker colour representing more significant p-values (see scale). In the cortex, cTBS induced a significant upregulation of processes involved 1.) metabolic processes with increased translation and ribosomal subunit assembly; and 2.) response to stimuli with increased cellular response to hydrogen peroxide.
(PDF)

**S5 Fig. Overrepresented gene ontologies in downregualted genes upon cTBS in the hippocampus.** Significantly overrepresented gene ontologies are in grey, with darker colour representing more significant p-values (see scale). In the cortex, cTBS induced a significant downregulation of processes involved 1.) negative regulators of transcription and translation; 2.) cellular response to brain-derived neurotrophic factor (BDNF); 3.) regulation of transforming growth factor beta receptor signalling pathway; and 4.) cellular development processes with decreased neuron differentiation.
(PDF)

**S6 Fig. In vitro stimulation of rat astroglial cell cultures by cTBS.** (**A**) Astroglial cells–containing astrocytes (GFAP), microglia and oligodendrocytes (not shown)–isolated from neonatal rat brains were kept in culture for 2 weeks before stimulation. (**B**) Stimulation with 4 trains of cTBS on two consecutive days increased cytokine release. Significantly increased release of IL-1β, IL-10 and TNF-α was found after stimulation with cTBS, which also increased levels of IL-6 but only with a statistical trend (p = 0.104). An unpaired Student t test was used to assess statistical differences between cTBS-stimulated and control cell cultures.
(PDF)

**S1 Table. Overrepresented gene ontologies in downregulated genes after cTBS in the cortex.**
(DOCX)

**S2 Table. Overrepresented gene ontologies in upregulated genes after cTBS in the cortex.**
(DOCX)

**S3 Table. Overrepresented gene ontologies in downregulated genes after cTBS in the hippocampus.**
(DOCX)

**S4 Table. Overrepresented gene ontologies in upregulated genes after cTBS in the hippocampus.**
(DOCX)

**S5 Table. Overrepresented gene ontologies in downregulated genes after iTBS in the cortex.**
(DOCX)

**S6 Table. Overrepresented gene ontologies in upregulated genes after iTBS in the cortex.**
(DOCX)

**S1 File.**
(ZIP)

## Acknowledgments

The authors thank PD Dario Cazzoli for his crucial contribution during previous experiments that laid the foundation to this study. Furthermore, we thank Franziska Simon, Sabrina Hupp, and Robert Lukesch for excellent technical support. We would like to thank the Next Generation Sequencing Platform of the University of Bern for performing the high-throughput sequencing experiments. Productive discussions and inputs from the ESCMID Study Group for Infectious Diseases of the Brain (ESGIB) were highly appreciated.

## Author Contributions

**Conceptualization:** Lukas Muri, Denis Grandgirard, Rémy Bruggmann, René M. Müri, Stephen L. Leib.

**Data curation:** Lukas Muri, Simone Oberhänsli, Michelle Buri, Ngoc Dung Le, Denis Grandgirard.

**Formal analysis:** Lukas Muri, Simone Oberhänsli, Denis Grandgirard, Rémy Bruggmann, René M. Müri, Stephen L. Leib.

**Funding acquisition:** René M. Müri, Stephen L. Leib.

**Investigation:** Lukas Muri, Simone Oberhänsli, Michelle Buri, Ngoc Dung Le, Denis Grandgirard, Rémy Bruggmann, René M. Müri, Stephen L. Leib.

**Methodology:** Lukas Muri, Simone Oberhänsli, Denis Grandgirard, Rémy Bruggmann, René M. Müri, Stephen L. Leib.

**Project administration:** René M. Müri, Stephen L. Leib.

**Resources:** Rémy Bruggmann, René M. Müri, Stephen L. Leib.

**Software:** Simone Oberhänsli, Rémy Bruggmann.

**Supervision:** Rémy Bruggmann, René M. Müri, Stephen L. Leib.

**Validation:** Lukas Muri, Simone Oberhänsli, Denis Grandgirard, Rémy Bruggmann, René M. Müri, Stephen L. Leib.

**Visualization:** Lukas Muri, Simone Oberhänsli.

**Writing – original draft:** Lukas Muri.

**Writing – review & editing:** Lukas Muri, Simone Oberhänsli, Michelle Buri, Ngoc Dung Le, Denis Grandgirard, Rémy Bruggmann, René M. Müri, Stephen L. Leib.

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
