## [Decision Letter · Decision Letter 0]

18 May 2020

PONE-D-20-11515

Repetitive Transcranial Magnetic Stimulation Activates Glial Cells and Inhibits Neurogenesis after Pneumococcal Meningitis

PLOS ONE

Dear Dr. Leib,

Thank you for submitting your manuscript to PLOS ONE. After careful consideration, we feel that it has merit but does not fully meet PLOS ONE’s publication criteria as it currently stands. Therefore, we invite you to submit a revised version of the manuscript that addresses the points raised during the review process. In addition to the reviewers' comments, I recommend a revision of statistical analysis because results are presented in non-standard manner, as you stated that they are illustrated by "mean values ± 95% confidence interval" instead of the commonly used standard deviation. The confidence interval is normally used when median values instead of mean values are illustrated. For this reason, please make clear that your data were normally distributed. You used a 2-way analysis of variance for multiple data. In this case, results of interaction between the main factors should also be reported. Note also that F values and degrees of freedom were omitted. Finally, statistical methods should be detailed also in legends.

We would appreciate receiving your revised manuscript for Jul 02 2020 11:59PM. To enhance the reproducibility of your results, we recommend that if applicable you deposit your laboratory protocols in protocols.io, where a protocol can be assigned its own identifier (DOI) such that it can be cited independently in the future. For instructions see: http://journals.plos.org/plosone/s/submission-guidelines#loc-laboratory-protocols

We look forward to receiving your revised manuscript.

Kind regards,

Giuseppe Biagini, MD

Academic Editor

PLOS ONE

3. Your ethics statement must appear in the Methods section of your manuscript. If your ethics statement is written in any section besides the Methods, please move it to the Methods section and delete it from any other section. Please also ensure that your ethics statement is included in your manuscript, as the ethics section of your online submission will not be published alongside your manuscript.

Reviewers' comments:

Reviewer's Responses to Questions

**Comments to the Author**

1. Is the manuscript technically sound, and do the data support the conclusions?

Reviewer #1: Yes

Reviewer #2: Yes

2. Has the statistical analysis been performed appropriately and rigorously? 

Reviewer #1: Yes

Reviewer #2: Yes

3. Have the authors made all data underlying the findings in their manuscript fully available?

Reviewer #1: Yes

Reviewer #2: Yes

4. Is the manuscript presented in an intelligible fashion and written in standard English?

Reviewer #1: Yes

Reviewer #2: Yes

5. Review Comments to the Author

Reviewer #1: The paper investigates the effects of repetitive transcranial magnetic stimulation on infant rats, that concomitantly cured from a pneumococcal meningitis. Two protocols of treatment have been used - continuous vs intermittent theta burst stimulation – and compared to each other in terms local gene expression, by whole transcriptome analysis, and glial marker modulation, by immunofluorescence analysis. Both cortex and hippocampus areas have been assessed in parallel. Finally, primary glial cell cultures exposed to magnetic stimulation in vitro have been evaluated for cytokine release in order to establish their degree of response.

The rational of the study has been correctly posed; the methodology is appropriate and data analyses have been adequately performed; results have been carefully described.

As mentioned by the Authors at the end of the discussion, the most relevant limitations of the study are the little number of animals tested, the lack of the untreated group and the one-shot analysis.

Nevertheless, the approach is innovative and results provide novel information. Overall, a huge amount of work has been done, just because of the very essential experimental protocol.

Reviewer #2: This manuscript by Lukas et al. suggests that cTBS intensified neuroinflammation after PM, which translated into increased release of pro-inflammatory mediators thereby inhibiting neuroregeneration.

Authors obtained quite interesting findings; however, the following points need to be deeply discussed:

- Lines 37-39: “During pneumococcal meningitis (PM), bacterial proliferation and autolysis in the cerebrospinal fluid (CSF) causes an excessive inflammatory reaction, which is associated with blood brain-barrier (BBB) breakdown, increased intracranial pressure, hydrocephalus and cerebral ischemia [1].” Do you know whether the specific rTMS protocols have some effects on the BBB breakdown in your animal model? Could structural and functional integrity of the BBB be modified by these protocols? Do you know whether the assessment of in vivo permeability to Evans blue and the quantification of the immunoreactivity to tight junction proteins (Vinet et al., 2018; Rincel et al., 2019) after the specific rTMS protocols were determined in your animal model? It would be interesting to further discuss these aspects in the discussion.

- Lines 71-74: “Intermittent or continuous theta burst stimulation (iTBS or cTBS) represent specific and very potent rTMS protocols during which stimulations are applied as bursts of 3-5 pulses at 30-100 Hz repeated at 5 Hz, with iTBS lowering cortical excitability and cTBS enhancing it [29].” Do you know whether these protocols could also be used in animal model of status epilepticus and epilepsy affected by changes in brain oscillations (Phelan et al., 2017; Costa et al., 2020)?

- Why did you perform the stimulations five days after the injection (lines 101-102)?

- The limited number of rats per group and the lack of two important groups (i.e., an uninfected control group receiving the same stimulation procedures and a group experiencing a long-term TBS exposure) could be crucial limitations of the study, and they should be deeply discussed.

References:

1. Vinet et al. (2018) A hydroxypyrone-based inhibitor of metalloproteinase-12 displays neuroprotective properties in both status epilepticus and optic nerve crush animal models. International Journal of Molecular Sciences. doi: 10.3390/ijms19082178

2. Rincel et al. (2019) Pharmacological restoration of gut barrier function in stressed neonates partially reverses long-term alterations associated with maternal separation. Psychopharmacology. doi: 10.1007/s00213-019-05252-w

3. Phelan et al. (2017) TRPC3 channels play a critical role in the theta component of pilocarpine-induced Status Epilepticus in mice. Epilepsia. doi:10.1111/epi.13648

4. Costa et al. (2020) Status epilepticus dynamics predicts latency to spontaneous seizures in the kainic acid model. Cell Physiol Biochem. doi: 10.33594/000000232

6. PLOS authors have the option to publish the peer review history of their article (what does this mean?). If published, this will include your full peer review and any attached files.

Reviewer #1: No

Reviewer #2: No

---

## [Author Response · Author response to Decision Letter 0]

2 Jul 2020

Response to Editor and Reviewers

Repetitive Transcranial Magnetic Stimulation Activates Glial Cells and Inhibits Neurogenesis after Pneumococcal Meningitis

Lukas Muri, Simone Oberhänsli, Michelle Buri, Ngoc Dung Le, Denis Grandgirard, Rémy Bruggmann, René M. Müri and Stephen L. Leib

PLOS ONE

Editor’s comment:

Dear Dr. Leib,

Thank you for submitting your manuscript to PLOS ONE. After careful consideration, we feel that it has merit but does not fully meet PLOS ONE’s publication criteria as it currently stands. Therefore, we invite you to submit a revised version of the manuscript that addresses the points raised during the review process. In addition to the reviewers' comments, I recommend a revision of statistical analysis because results are presented in non-standard manner, as you stated that they are illustrated by "mean values ± 95% confidence interval" instead of the commonly used standard deviation. The confidence interval is normally used when median values instead of mean values are illustrated. For this reason, please make clear that your data were normally distributed. You used a 2-way analysis of variance for multiple data. In this case, results of interaction between the main factors should also be reported. Note also that F values and degrees of freedom were omitted. Finally, statistical methods should be detailed also in legends.

Response and measures taken:

We are thankful for the editor’s comment to point out these suggestions for improvement within our statistical appraisal.

The only graph, where confidence intervals (CI) instead of standard deviations (SD) were presented, was Fig. S1 with percentual weight change of investigated animals. This graph was changed accordingly and shows now SDs. The same graph investigates weight change over time with a 2-way ANOVA, for which we further included numbers on interaction, F-values and degrees of freedom as suggested by the editor. Generally, statistical methods were added to figure legends.

The following changes (marked in yellow) were implemented in the manuscript:

Methods:

Results are presented as mean values ± standard deviation (SD) if not stated otherwise. (Lines 180-181)

Results:

After infection, no statistically significant difference between sham and iTBS (2-way ANOVA F=0.2059 and p=0.6524 with degree of freedom (df) = 1; interaction of main factors p=0.9866) or sham and cTBS (2-way ANOVA F=0.0004 and p=0.9847 with df=1; interaction of main factors p=0.9999) in terms of weight development was detectable (Fig. S1B). (Lines 190-193)

Figure legends:

Figure 3. cTBS increases abundance of CD68+ cells in cerebral cortex and hippocampal dentate gyrus. Exposure to cTBS – but not to iTBS – increased the number of CD68+ cells in the granular cell layer of the hippocampal dentate gyrus (DG) compared to sham-stimulated and iTBS-stimulated animals (A,C). Similarly, cTBS increased the abundance of CD68+ cells in the cerebral cortex (B,D). Statistical differences were assessed using one-way ANOVA with Tukey’s multiple comparison test to adjust for multiple testing. (Lines 288-293)

Figure S1. Bacterial CSF titres and weight development during pneumococcal meningitis. Bacterial CSF titres were comparable between the three different groups cTBS, iTBS and sham and indicated a comparable severity of infection (A). Development of relative weight after infection further proved comparability between different groups, as PM-induced weight loss and recovery was non-different within all analysed animals analysed by 2-way ANOVA (B). Statistical differences for bacterial titres were assessed using one-way ANOVA with Tukey’s multiple comparison test to adjust for multiple testing. (Lines 667-673)

Figure S6. In vitro stimulation of rat astroglial cell cultures by cTBS. (A) Astroglial cells – containing astrocytes (GFAP), microglia and oligodendrocytes (not shown) – isolated from neonatal rat brains were kept in culture for 2 weeks before stimulation. (B) Stimulation with 4 trains of cTBS on two consecutive days increased cytokine release. Significantly increased release of IL-1β, IL-10 and TNF-α was found after stimulation with cTBS, which also increased levels of IL-6 but only with a statistical trend (p = 0.104). An unpaired Student t test was used to assess statistical differences between cTBS-stimulated and control cell cultures. (Lines 703-709)

 

Reviewers' comments:

Reviewer #1: 

The paper investigates the effects of repetitive transcranial magnetic stimulation on infant rats, that concomitantly cured from a pneumococcal meningitis. Two protocols of treatment have been used - continuous vs intermittent theta burst stimulation – and compared to each other in terms local gene expression, by whole transcriptome analysis, and glial marker modulation, by immunofluorescence analysis. Both cortex and hippocampus areas have been assessed in parallel. Finally, primary glial cell cultures exposed to magnetic stimulation in vitro have been evaluated for cytokine release in order to establish their degree of response.

The rational of the study has been correctly posed; the methodology is appropriate and data analyses have been adequately performed; results have been carefully described.

As mentioned by the Authors at the end of the discussion, the most relevant limitations of the study are the little number of animals tested, the lack of the untreated group and the one-shot analysis.

Nevertheless, the approach is innovative and results provide novel information. Overall, a huge amount of work has been done, just because of the very essential experimental protocol.

Response to reviewer:

We thank the reviewer for this very positive feedback and are happy to see that our experiments are judged to be methodologically appropriate. Thank you for reviewing and appreciating our manuscript.

 

Reviewer #2: 

This manuscript by Lukas et al. suggests that cTBS intensified neuroinflammation after PM, which translated into increased release of pro-inflammatory mediators thereby inhibiting neuroregeneration.

Authors obtained quite interesting findings; however, the following points need to be deeply discussed:

- Lines 37-39: “During pneumococcal meningitis (PM), bacterial proliferation and autolysis in the cerebrospinal fluid (CSF) causes an excessive inflammatory reaction, which is associated with blood brain-barrier (BBB) breakdown, increased intracranial pressure, hydrocephalus and cerebral ischemia [1].” 

Do you know whether the specific rTMS protocols have some effects on the BBB breakdown in your animal model? Could structural and functional integrity of the BBB be modified by these protocols? Do you know whether the assessment of in vivo permeability to Evans blue and the quantification of the immunoreactivity to tight junction proteins (Vinet et al., 2018; Rincel et al., 2019) after the specific rTMS protocols were determined in your animal model? It would be interesting to further discuss these aspects in the discussion.

Response and measures taken:

We thank the reviewer for identifying this important point about rTMS and BBB opening. Generally, high-intensity magnetic stimulation harbours the potential to increase BBB permeability most likely via increased release of glutamate [1]. Experimental and human data suggest that TMS induces BBB opening and facilitates delivery of drugs to neural tissue [1]. Vazana et al. further conclude that TMS protocols may represent a future approach to non-invasively induce a transient BBB opening for improved drug delivery of chemotherapeutics into the CNS [1]. Unfortunately, the basic mechanisms of the theta burst stimulation (TBS) protocols used within our study are not studied in detail until now. We are not aware about studies dealing with in vivo permeability to Evans blue and the quantification of the immunoreactivity to tight junction proteins using these specific TBS protocols in the described Wistar rat pneumococcal meningitis model. 

Nevertheless, as stated by the reviewer, TMS protocols harbour the potential to change BBB permeability. Additionally, increased brain endothelial permeability to serum proteins was found to be associated with astrocyte transformation with increased GFAP levels and neuroinflammation [2–5]. Increased BBB permeability upon cTBS treatment in our study might represent an additional explanation for elevated neuroinflammation markers and might further explain why we only observed low increases in in vitro experiments focussing on primary astroglial cell cultures. 

In response to the point risen by the reviewer, the following changes (marked in yellow) were implemented in the discussion of the manuscript:

Yet another explanation for increased neuroinflammation in vivo might arise from rTMS-induced changes in BBB permeability. High-intensity magnetic stimulation harbours the potential to increase BBB permeability most likely via increased release of glutamate [60]. Increased brain endothelial permeability to serum proteins was found to be associated with neuroinflammatory markers such as astrocyte transformation showing increased GFAP levels [61–64]. Changes in BBB permeability upon TBS and direct activation of glial cells might both contribute to the observed neuroinflammatory reaction upon magnetic stimulation. More experiments are needed to confirm these hypotheses. (Lines 356-363)

 

- Lines 71-74: “Intermittent or continuous theta burst stimulation (iTBS or cTBS) represent specific and very potent rTMS protocols during which stimulations are applied as bursts of 3-5 pulses at 30-100 Hz repeated at 5 Hz, with iTBS lowering cortical excitability and cTBS enhancing it [29].” 

Do you know whether these protocols could also be used in animal model of status epilepticus and epilepsy affected by changes in brain oscillations (Phelan et al., 2017; Costa et al., 2020)?

Response and measures taken:

To the best of our knowledge, there are no published studies concerning the applications of TBS in animal models of status epilepticus. However, there is one publication assessing the suppression of acute 4-aminopyridine-induced seizures by electrical TBS (eTBS) from 2014 [6]. The authors found that eTBS suppressed induced seizures depending on stimulation settings and concluded that eTBS might represent a novel protocol for seizure treatment. Nevertheless, electrical stimulation cannot be compared to magnetic stimulations and these results are therefore hard to correlate to magnetic TBS protocols. Furthermore, we found a conference abstract about the effect of cTBS in four human subjects with refractory neocortical epilepsy [7]. The authors also state that there is very limited data on TBS in the context of epilepsy. Furthermore, they found in their small pilot study that cTBS in epileptic patients is safe and well-tolerated and might show potential to reduce seizure frequency [7]. 

- Why did you perform the stimulations five days after the injection (lines 101-102)?

Response and measures taken:

We are glad that the reviewer raised this question and realised that we need to further clarify this in the manuscript. 

The intention of this study was to assess the potential of TBS protocols in provoking neuroregeneration after pneumococcal meningitis. Our rational was to start TBS as soon as animals were microbiologically cured from pneumococcal infection. From published data on our infant pneumococcal meningitis model, we know that already six hours after antibiotic therapy, bacterial loads in CSF are massively diminished with sterile CSF observed in many animals [8,9]. Inflammatory cytokines raise during acute infection but return mostly to baseline levels within 24 hours after treatment initiation [8–11]. Clinical scoring further reveals recovery within 24 hours after antibiotic therapy start in different models of experimental pneumococcal meningitis [9,12]. While infected animals lose weight during acute infection, they recover and start to gain wait starting from day 3 after infection [11] indicating cure from infection and the start of the recovery phase. In a mouse model of pneumococcal meningitis, leukocyte counts in CSF increased to high levels during acute meningitis but were tremendously reduced within two days after antibiotic therapy [13]. Finally, a previous transcriptomic analysis revealed that the major events in the regulation of the host response on a transcriptional level occur within the first 3 days after infection [14]. Taking all these findings together, we can be certain that at day 5 after infection, rats are microbiologically cured from pneumococcal infection and infection-induced neuroinflammation with infiltrating leukocytes should be back to baseline levels. Thereby we do not risk to interfere with neuroinflammatory processes during acute infection and maximise treatment potential with starting as early as possible.

The following adjustments (marked in yellow) were made in the text to clearly state the rationale behind starting stimulations at day 5 after infection:

Methods:

Five days after infection – when the acute phase of infection with associated neuroinflammation is overcome and animals start to recover from pneumococcal meningitis – animals were randomized to receive continuous theta burst stimulation … (Lines 97-99)

Discussion:

In the present study, we studied the impact of TBS in infant rats microbiologically cured from PM after five days of antibiotic therapy. To ensure that magnetic stimulation does not interfere with neuroinflammatory processes during acute infection, stimulation was initiated at day five after infection. Previous studies revealed that at this time animals successfully survived PM and start to recover with neuroinflammatory markers and clinical scores being back at baseline and animals starting to gain weight again [38,39,41–44]. (Lines 303-308).

- The limited number of rats per group and the lack of two important groups (i.e., an uninfected control group receiving the same stimulation procedures and a group experiencing a long-term TBS exposure) could be crucial limitations of the study, and they should be deeply discussed.

Response and measures taken:

We completely agree with the reviewer that these are the biggest limitations of our study. We already mentioned these points in our discussion but have now extended their discussion as follows:

“Our study has some limitations, including small numbers of animals in the in vivo experiments and the lack of an uninfected control group receiving the same stimulation procedures. As we were primarily interested in investigating the effects of TBS after PM and did not intent to focus on differences in gene expression between infected and non-infected animals, this control group was omitted. The effect of TBS on differential gene expression in healthy animals would represent another story. Furthermore, it would have been interesting to also analyse the effect of long-term TBS exposure on neurogenesis and neuroinflammation. However, this would have been beyond the scope of this study focussed on differential gene expression induced by TBS after acute PM.” (Lines 403-410)

 

References:

1 Vazana U, Veksler R, Pell GS, Prager O, Fassler M, Chassidim Y, et al. Glutamate-mediated blood–brain barrier opening: Implications for neuroprotection and drug delivery. J Neurosci 2016; 36:7727–7739.

2 Seiffert E, Dreier JP, Ivens S, Bechmann I, Tomkins O, Heinemann U, et al. Lasting blood-brain barrier disruption induces epileptic focus in the rat somatosensory cortex. J Neurosci 2004; 24:7829–7836.

3 Ivens S, Kaufer D, Flores LP, Bechmann I, Zumsteg D, Tomkins O, et al. TGF-b receptor-mediated albumin uptake into astrocytes is involved in neocortical epileptogenesis. Brain 2007; 130:535–547.

4 David Y, Cacheaux LP, Ivens S, Lapilover E, Heinemann U, Kaufer D, et al. Astrocytic dysfunction in epileptogenesis: Consequence of altered potassium and glutamate homeostasis? J Neurosci 2009; 29:10588–10599.

5 Weissberg I, Wood L, Kamintsky L, Vazquez O, Milikovsky DZ, Alexander A, et al. Albumin induces excitatory synaptogenesis through astrocytic TGF-β/ALK5 signaling in a model of acquired epilepsy following blood-brain barrier dysfunction. Neurobiol Dis 2015; 78:115–125.

6 Siah BH, Chiang CC, Ju MS, Lin CCK. Suppression of acute seizures by theta burst electrical stimulation of the hippocampal commissure using a closed-loop system. Brain Res 2014; 1593:117–125.

7 Carrette S, Klooster D, Staljanssens W, Van Mierlo P, Van Dycke A, Carrette E, et al. Continuous thetaburst stimulation for the treatment of refractory neocortical epilepsy. Front Neurosci Conf Abstr 12th Natl Congr Belgian Soc Neurosci 2017; 11. doi:10.3389/conf.fnins.2017.94.00073

8 Muri L, Perny M, Zemp J, Grandgirard D, Leib SL. Combining Ceftriaxone with Doxycycline and Daptomycin Reduces Mortality, Neuroinflammation, Brain Damage and Hearing Loss in Infant Rat Pneumococcal Meningitis. Antimicrob Agents Chemother Published Online First: 6 May 2019. doi:10.1128/AAC.00220-19

9 Muri L, Grandgirard D, Buri M, Perny M, Leib SL. Combined effect of non-bacteriolytic antibiotic and inhibition of matrix metalloproteinases prevents brain injury and preserves learning, memory and hearing function in experimental paediatric pneumococcal meningitis. J Neuroinflammation 2018; 15:233.

10 Erni ST, Fernandes G, Buri M, Perny M, Rutten RJ, van Noort JM, et al. Anti-inflammatory and Oto-Protective Effect of the Small Heat Shock Protein Alpha B-Crystallin (HspB5) in Experimental Pneumococcal Meningitis. Front Neurol 2019; 10:570.

11 Muri L, Le ND, Zemp J, Grandgirard D, Leib SL. Metformin mediates neuroprotection and attenuates hearing loss in experimental pneumococcal meningitis. J Neuroinflammation 2019; 16:156.

12 Klein M, Höhne C, Angele B, Högen T, Pfister HW, Tüfekci H, et al. Adjuvant non-bacteriolytic and anti-inflammatory combination therapy in pneumococcal meningitis: an investigation in a mouse model. Clin Microbiol Infect Published Online First: 9 April 2018. doi:10.1016/j.cmi.2018.03.039

13 Koedel U, Frankenberg T, Kirschnek S, Obermaier B, Häcker H, Paul R, et al. Apoptosis Is Essential for Neutrophil Functional Shutdown and Determines Tissue Damage in Experimental Pneumococcal Meningitis. PLoS Pathog 2009; 5:e1000461.

14 Wittwer M, Grandgirard D, Rohrbach J, Leib SL. Tracking the transcriptional host response from the acute to the regenerative phase of experimental pneumococcal meningitis. BMC Infect Dis 2010; 10:176.

---

## [Decision Letter · Decision Letter 1]

22 Jul 2020

Repetitive Transcranial Magnetic Stimulation Activates Glial Cells and Inhibits Neurogenesis after Pneumococcal Meningitis

PONE-D-20-11515R1

Dear Dr. Leib,

We’re pleased to inform you that your manuscript has been judged scientifically suitable for publication and will be formally accepted for publication once it meets all outstanding technical requirements.

Kind regards,

Giuseppe Biagini, MD

Academic Editor

PLOS ONE

Additional Editor Comments (optional):

Reviewers' comments:

Reviewer's Responses to Questions

**Comments to the Author**

1. If the authors have adequately addressed your comments raised in a previous round of review and you feel that this manuscript is now acceptable for publication, you may indicate that here to bypass the “Comments to the Author” section, enter your conflict of interest statement in the “Confidential to Editor” section, and submit your "Accept" recommendation.

Reviewer #1: All comments have been addressed

Reviewer #2: All comments have been addressed

2. Is the manuscript technically sound, and do the data support the conclusions?

Reviewer #1: Yes

Reviewer #2: (No Response)

3. Has the statistical analysis been performed appropriately and rigorously? 

Reviewer #1: Yes

Reviewer #2: (No Response)

4. Have the authors made all data underlying the findings in their manuscript fully available?

Reviewer #1: Yes

Reviewer #2: (No Response)

5. Is the manuscript presented in an intelligible fashion and written in standard English?

Reviewer #1: Yes

Reviewer #2: (No Response)

6. Review Comments to the Author

Reviewer #1: the paper has been adequately upgraded. the points raised by the referee have been carefully considered

Reviewer #2: (No Response)

7. PLOS authors have the option to publish the peer review history of their article (what does this mean?). If published, this will include your full peer review and any attached files.

Reviewer #1: No

Reviewer #2: No

---

## [Editor Report · Acceptance letter]

21 Aug 2020

PONE-D-20-11515R1 

Repetitive transcranial magnetic stimulation activates glial cells and inhibits neurogenesis after pneumococcal meningitis 

Dear Dr. Leib:

I'm pleased to inform you that your manuscript has been deemed suitable for publication in PLOS ONE. Congratulations! Your manuscript is now with our production department. 

Kind regards, 

on behalf of

Dr. Giuseppe Biagini 

Academic Editor

PLOS ONE